# Development of the BioHybrid Assay: Combining Primary Human Vascular Smooth Muscle Cells and Blood to Measure Vascular Calcification Propensity

**DOI:** 10.3390/cells10082097

**Published:** 2021-08-16

**Authors:** Armand M. G. Jaminon, Asim C. Akbulut, Niko Rapp, Rafael Kramann, Erik A. L. Biessen, Lieve Temmerman, Barend Mees, Vincent Brandenburg, Robert Dzhanaev, Willi Jahnen-Dechent, Juergen Floege, Jouni Uitto, Chris P. Reutelingsperger, Leon J. Schurgers

**Affiliations:** 1Department of Biochemistry, Cardiovascular Research Institute Maastricht, Maastricht University Medical Centre, 6200 MD Maastricht, The Netherlands; a.jaminon@maastrichtuniversity.nl (A.M.G.J.); a.akbulut@maastrichtuniversity.nl (A.C.A.); n.rapp@maastrichtuniversity.nl (N.R.); c.reutelingsperger@maastrichtuniversity.nl (C.P.R.); 2Institute of Experimental Medicine and Systems Biology, RWTH Aachen University, 52074 Aachen, Germany; rkramann@ukaachen.de; 3Department of Nephrology and Clinical Immunology, RWTH Aachen University Hospital, 52074 Aachen, Germany; jfloege@ukaachen.de; 4Department of Pathology, Cardiovascular Research Institute Maastricht, Maastricht University, 6229 HX Maastricht, The Netherlands; erik.biessen@mumc.nl (E.A.L.B.); lieve.temmerman@mumc.nl (L.T.); 5Institute for Molecular Cardiovascular Research (IMCAR), Universitätsklinikum Aachen, 52074 Aachen, Germany; 6Department of Vascular Surgery, Maastricht University Medical Centre, 6229 HX Maastricht, The Netherlands; barend.mees@mumc.nl; 7Department of Cardiology, Rhein-Maas-Klinikum Würselen, 52146 Würselen, Germany; vincent.brandenburg@post.rwth-aachen.de; 8Helmholtz Institute for Biomedical Engineering, Biointerface Group, RWTH Aachen University, 52074 Aachen, Germany; rdzhanaev@ukaachen.de (R.D.); willi.jahnen@rwth-aachen.de (W.J.-D.); 9Department of Dermatology and Cutaneous Biology, Sidney Kimmel Medical College, Jefferson Institute of Molecular Medicine, Thomas Jefferson University, Philadelphia, PA 19107, USA; Jouni.Uitto@jefferson.edu

**Keywords:** vascular calcification, vascular smooth muscle cells, BioHybrid, fetuin-A, matrix Gla protein, vitamin K, vitamin K antagonist, cardiovascular disease

## Abstract

Background: Vascular calcification is an active process that increases cardiovascular disease (CVD) risk. There is still no consensus on an appropriate biomarker for vascular calcification. We reasoned that the biomarker for vascular calcification is the collection of all blood components that can be sensed and integrated into a calcification response by human vascular smooth muscle cells (hVSMCs). Methods: We developed a new cell-based high-content assay, the BioHybrid assay, to measure in vitro calcification. The BioHybrid assay was compared with the o-Cresolphthalein assay and the T50 assay. Serum and plasma were derived from different cohort studies including chronic kidney disease (CKD) stages III, IV, V and VD (on dialysis), pseudoxanthoma elasticum (PXE) and other cardiovascular diseases including serum from participants with mild and extensive coronary artery calcification (CAC). hVSMCs were exposed to serum and plasma samples, and in vitro calcification was measured using AlexaFluor^®^-546 tagged fetuin-A as calcification sensor. Results: The BioHybrid assay measured the kinetics of calcification in contrast to the endpoint o-Cresolphthalein assay. The BioHybrid assay was more sensitive to pick up differences in calcification propensity than the T50 assay as determined by measuring control as well as pre- and post-dialysis serum samples of CKD patients. The BioHybrid response increased with CKD severity. Further, the BioHybrid assay discriminated between calcification propensity of individuals with a high CAC index and individuals with a low CAC index. Patients with PXE had an increased calcification response in the BioHybrid assay as compared to both spouse and control plasma samples. Finally, vitamin K1 supplementation showed lower in vitro calcification, reflecting changes in delta Agatston scores. Lower progression within the BioHybrid and on Agatston scores was accompanied by lower dephosphorylated-uncarboxylated matrix Gla protein levels. Conclusion: The BioHybrid assay is a novel approach to determine the vascular calcification propensity of an individual and thus may add to personalised risk assessment for CVD.

## 1. Introduction

Vascular calcification is an active process that occurs within the vessel wall and increases the risk of cardiovascular disease (CVD) [1,2,3]. Moreover, coronary artery calcification (CAC) is considered a marker for cardiovascular burden. High abundance of calcification present at any vascular site relates to approximately a 3.5-fold increase for CVD events [4]. Additionally, CAC progression is associated with a 17-fold increased risk of myocardial infarction [5]. Therefore, the means to measure vascular calcification amount or progression rate have the potential to accurately predict the risk for cardiovascular events.

The current methods to detect and assess calcification in vivo aim to visualise precipitated calcium phosphate deposition in the vasculature. Circulating biomarkers that reflect the propensity to develop and aggravate vascular calcification have not been identified so far. In vitro, diagnostics by the T50 assay may infer calcification propensity in serum [6]. Given this, the T50 assay is based on a chemical reaction and is largely affected in samples wherein a mineral imbalance or decreased circulating fetuin-A or albumin is present, thus lacking biological cause or consequence. In this paper, we reason that the circulating biomarker for vascular calcification is not a single component but the collection of all components in circulation.

As readout of this composite biomarker, we selected in vitro calcification mediated by human vascular smooth muscle cells (hVSMCs), considered by many as the protagonists in vascular calcification [3]. This paper describes the BioHybrid assay, a cell-based assay for measuring the composite biomarker of vascular calcification. We measured the BioHybrid response that we consider to be the in vitro development of calcification. This was performed using either serum or plasma samples from several cohort studies. Cohorts included were that of variant stages of CKD wherein vascular calcification is synonymous with disease progression as well as PXE, a genetic disorder wherein ectopic mineralisation of soft tissue is present. In addition, we studied two cohort studies wherein enrolment was based on CVD status: one high vs. low CAC (determined by Agatston scoring), and the other was aortic valve calcification (AVC) with CAC. From both CVD cohorts, further information on Agatston scoring data was available.

In the following, we demonstrate that there is a strong and significant correlation between the Agatston score and the BioHybrid calcification readout. Additionally, serum and plasma samples from the various cohorts responded with increased in vitro calcification in the BioHybrid assay compared to respective controls. Serum or plasma samples from participants with extensive calcification responded with a faster development of in vitro hVSMC calcification than those with milder calcification. The BioHybrid calcification assay presented in this paper is an informative strategy to determine an individual’s vascular calcification propensity.

## 2. Materials and Methods

### 2.1. hVSMC Culturing and Characterisation

hVSMCs were isolated from non-atherosclerotic abdominal aortas of surgical biopsies in accordance with MUMC+ research and diagnostic procedure. Collection, storage and use of tissue and patient data were performed in agreement with the Dutch Code for Proper Secondary Use of Human Tissue (https://www.federa.org/codes-conduct accessed on 30 July 2021). This study complies with the Declaration of Helsinki.

Briefly, human vascular tissue samples were washed in phosphate buffered saline (PBS) before intima, fat and connective tissue was removed. Tissue was then cut into small fragments (approx. 2–5 mm in diameter) and left in M199 medium (Gibco, Bleiswijk, the Netherlands) containing 20% fetal bovine serum (FBS), 1% penicillin/streptomycin (PS, Gibco, Bleiswijk, the Netherlands) and 1% Amphotericin B on laminin coated plates (#L2020, Sigma, Saint Louis, MO, USA). When outgrowing cells reached confluency, they were passaged 1:2 on laminin-coated plates for two further passages. hVSMCs were routinely cultured in M199 medium with 20% FBS and 1% PS. For experiments, cells were used between passages 5–10. All samples were mycoplasma tested.

For characterisation (Appendix A), hVSMCs were fixed in 4% paraformaldehyde before blocking in 2% BSA, 0.1% triton in PBS for one hour. Primary antibodies were incubated overnight at 4 °C: alpha-Smooth Muscle Actin (αSMA) 1:200 (DAKO, Glostrup, Glostrup, Denmark, M0851), phosphorylated-Myosin Light Chain (pMLC) 1:200 (Cell Signaling, Danvers, MA, USA, 3675S), Calponin (CNN1) 1:200 (Abcam, Cambridge, UK, ab46794), Smooth Muscle protein 22-alpha (SM22a) 1:200 (Abcam, Cambridge, UK ab14106) and S100 calcium-binding protein A4 (S100A4) 1:200 (DAKO, Glostrup, Glostrup, Denmark, A5114). Cells were incubated with secondary antibodies for one hour at room temperature: anti-mouse FITC 1:250 (Hycult Biotech, Uden, The Netherlands, HP2001) or anti-rabbit FITC 1:250 (DAKO, Glostrup, Glostrup, Denmark, F0205). Before imaging, cells were incubated with 4′,6-diamidino-2-phenylindole (DAPI) for one minute. Imaging was performed on the Cytation 3 system on 20× magnification (BioSPX, Abcoude, The Netherlands).

### 2.2. BioHybrid Assay

For calcification experiments, hVSMCs were seeded in culture well plates at a density of 10–15 × 10^3^ cells/cm^2^. After 24 h, hVSMCs were cultured in calcification medium (M199, 5% human serum or plasma, 1% PS and 3.6 mM Ca^2+^) for up to 14 days. Plasma samples were further supplemented with 143 µM Hirudin to prevent coagulation upon re-calcifying plasma. As positive control, a 0.5% serum/plasma condition was used due to the higher magnitude of calcification. For the BioHybrid assay, fetuin-A-AlexaFluor^®^-546 (1–3 µg/mL; prepared in-house) and Hoechst 33,342 (1 µg/mL, Invitrogen, Waltham, MA, USA) were supplemented at the start of the experiment. At various time points after calcification induction, red fluorescent protein (RFP) and DAPI channel (cell count) were imaged up to 14 days. After calcification was detected, cells were imaged regularly to follow calcification progression. Imaging was done with the Cytation 3 system (BioSPX, Abcoude, the Netherlands) and analysed using Gen5 software v.2.9 (BioTek, Abcoude, the Netherlands). As readout RFP confluence per well was normalised against cell count.

### 2.3. o-Cresolphthalein Assay

The o-Cresolphthalein assay (Randox, London, UK) was carried out according to manufacturer’s instructions. In brief, cells were washed twice in PBS and mineral deposits were solubilised in 0.1 M HCl for 2 h. After mineral deposits were solubilised, o-Cresolphthalein was added, which forms a violet complex with the calcium in the supernatant. A calcium standard was prepared with a range 0–2.54 mmol/L and concentration was determined by measuring absorbance at 570 nm.

### 2.4. T50 Assay

The T50 assay was performed as previously described [6]. In brief, sera of patients were centrifuged at 10,000 g and the supernatant was mixed with high concentrations of calcium and phosphate solutions to induce calciprotein particle (CPP) formation. Pipetting was performed with a high precision pipetting device (Liquidator, Mettler Toledo, Greifensee, Switzerland) and formation of CPP was monitored in a time-resolving manner using a standard nephelometer (Nephelostar, BMG Labtech, Ortenberg, Germany). The results were used to calculate the one-half maximal transition time, hence T50.

### 2.5. Serum and Plasma Preparation

Serum and EDTA plasma samples were obtained from several cohort studies. Enrolment to the cohorts was based on clinical diagnosis of the following parameters: CKD5D, CKD 3 to 5, PXE, high and low CAC score and AVC. In the AVC trial, patients received either vitamin K1 supplementation or placebo [7]. Serum and plasma of healthy individuals were used as a negative control. All patients and healthy volunteers had given written consent. This research and sample collection were approved by the Ethics Committee of the RWTH Aachen University Hospital, Aachen and Sidney Kimmel Medical College at Thomas Jefferson University, Philadelphia, Pennsylvania and conducted in accordance with the Declaration of Helsinki.

### 2.6. Dephosphorylated-Uncarboxylated Matrix Gla Protein (dp-ucMGP) Measurement

Plasma dp-ucMGP levels were determined using the commercially available IVD CE-marked chemiluminescent InaKtif MGP assay on the IDS-iSYS system (IDS, Boldon, UK). In brief, samples and internal calibrators were incubated with magnetic particles coated with murine monoclonal antibodies against dp-MGP, acridinium-labelled murine monoclonal antibodies against ucMGP and an assay buffer. The magnetic particles were captured and washed to remove unbound analyte. Trigger reagents were added; the resulting light emitted by the acridinium label is directly proportional to the level of dp-ucMGP in the sample. The within-run and total variations of this assay were 0.8–6.2% and 3.0–8.2%, respectively. The assay measuring range was between 300 and 12,000 pmol/L and was linear up to 11,651 pmol/L [8]. All assays were performed by Coagulation Profile laboratories, Maastricht, the Netherlands.

### 2.7. Data Analysis and Statistics

Real-time calcification development over time was analysed using Gen5 version 2.9 (BioTek, Abcoude, the Netherlands). Confluence of Alexa 546 signal was determined and normalised against cell count. Data are presented as mean ± standard deviation (SD). For over-time measurements, data is presented as mean. Non-parametric Mann−Whitney U test was performed for comparison between two groups. For more than two groups, significance was determined using one-way analysis of variance (ANOVA; Kruskal−Wallis) with comparison between groups by Dunn’s multiple comparison test. For correlation determination, a linear regression model and R2 was determined. Statistical significance was defined as *p* ≤ 0.05 (*), *p* ≤ 0.01 (**), *p* ≤ 0.001 (***) and *p* ≤ 0.0001 (****).

## 3. Results

### 3.1. Development of the BioHybrid Calcification Assay

Thus far, measurement of in vitro hVSMC calcification is only possible via end-point assays using either Alizarin Red or o-Cresolphthalein quantification. We developed a novel in vitro hVSMC calcification assay in which calcification development can be followed in real-time using fluorescently labelled fetuin-A-AlexaFluor^®^-546. To test the robustness of our live calcification assay, we compared the BioHybrid assay with the o-Cresolphthalein method, which measures calcification as calcium/protein ratio. We found a strong correlation (R^2^ = 0.72) between fetuin-A-AlexaFluor^®^-546 fluorescence per cell (% RFP/cell count) and o-Cresolphthalein (µg Ca^2+^/µg protein; Figure 1A; representative images Figure 1C; curve fit Appendix A). Moreover, the BioHybrid assay has a dynamic range and is significantly more sensitive in both the earlier and lower detection of in vitro calcification compared to the o-Cresolphthalein method (Figure 1B).

A great advantage of using fetuin-A-AlexaFluor^®^-546 is that it can be used in real-time, measuring in vitro hVSMC calcification progression over time. This provides sequential, quantifiable measurements of in vitro hVSMC calcification in one single assay. We tested a variety of both serum and calcium conditions to establish the optimal parameters for our assay. We found that the use of 5% serum or plasma was optimal for a robust increase in in vitro hVSMC calcification (Appendix A). Under 0.5% FBS condition, development of calcification was too rapid (Appendix A) and when using 10% FBS calcification formation was low, as shown by us previously [9]. Another parameter that is often not uniform in calcification assays is the final concentration of calcium to trigger calcification development. All our assays were performed with 3.6 mM of calcium chloride (CaCl_2_). Higher concentrations of CaCl_2_ (4.5 mM and 5.4 mM total calcium) led to a much faster development of calcification, comparable to the use of lower amounts of serum (Appendix A). Hence, we used 3.6 mM CaCl_2_ and 5% serum or plasma for our BioHybrid assay.

Since we developed the BioHybrid assay to test clinical samples, we investigated whether human serum or plasma behave similarly with respect to calcification development. Comparing FBS with either a pool of healthy control serum or plasma or the individual serum or plasma from each of our healthy controls, we did not find differences in hVSMC calcification as assessed using both the fetuin-A-AlexaFluor^®^-546 and the o-Cresolphthalein method (Figure 1D–E). This indicates that serum as well as plasma samples can be used to measure calcification with our assay and that the fetuin-A- AlexaFluor^®^-546 probe gives comparable results to that of the o-Cresolphthalein method.

### 3.2. Calcification Propensity of CKD5D Serum

First, we used the T50 assay to measure hVSMC calcification propensity in a variety of samples pre- and post-dialysis as well as control serum. Pre-dialysis serum produced a significantly quicker T50 value compared to a corresponding serum sample post-dialysis. The T50 value post-dialysis was similar to that of control samples (Figure 2A). When screening serum from the same samples in our BioHybrid assay, we confirmed a significant difference between pre- and post-dialysis serum after three days in the BioHybrid assay (Figure 2B). Additionally, we were able to note a significant difference between post-dialysis serum and control samples in the BioHybrid assay (Figure 2B).

After 6 days in the BioHybrid assay, pre- and post-dialysis serum continued to have progression of in vitro hVSMC calcification, whereas control samples remained low (Figure 2C). Moreover, serum from five out of the seven participants induced significantly less calcification post dialysis (Figure 2E). Serum from one individual had no significant difference and intriguingly, one showed a significant increase in calcification post-dialysis. This further suggests a sensitive and specific response that triggers in vitro hVSMC calcification and is only detectable by measuring real-time calcification development with the BioHybrid.

### 3.3. Calcification Propensity in Cohorts of Chronic Kidney Disease Patients, Pseudoxanthoma Elasticum Patients and Patients with Coronary Artery Calcification as Compared to Control

Next, we compared hVSMC calcification induction by serum of a cohort of participants with CKD stages 3, 4 or 5 against a pool of healthy donors. As shown in Figure 3A, healthy controls had the lowest rate of calcification development per hour. Serum from the CKD5 cohort participants showed the highest rate of hVSMC calcification followed by CKD4 and CKD3, respectively (trend; *p* = 0.0708). We observed a positive association of in vitro calcification rate from serum with increasing CKD severity (Figure 3B; *p* = 0.0859).

We also compared hVSMC calcification induced by plasma samples from a PXE cohort against plasma from spouse controls and plasma from healthy controls. We found that plasma from the PXE cohort induced in vitro calcification faster compared to both control and spouse plasma (Figure 3C). This difference was significant between the PXE group and spouse controls (Figure 3C; *p* = 0.0012).

Next, we assessed how the BioHybrid system responded to serum from individuals known to have either mild or extensive CAC as determined by Agatston scores. For this, we tested serum from individuals with either a high (>1500) or low (<50) Agatston score. In vitro calcification was significantly higher from serum of the high Agatston score group compared to the low Agatston score group (Figure 3D; *p* = 0.0022). High Agatston score patient samples displayed a significant linear regression with in vitro calcification (Figure 3E; R^2^ = 0.68) whereas no such correlation was observed in the low Agatston group (data not shown).

### 3.4. Calcification Propensity of Serum from Aortic Valve Calcification Patients with Vitamin K1 Treatment

Lastly, we tested serum samples from a cohort of individuals with AVC and CAC who participated in a proof-of-concept study with either vitamin K1 (phylloquinone) or placebo treatment for one year [7]. When comparing serum from the start of the trial with serum after 12 months of vitamin K1 supplementation, we found a reduction of in vitro hVSMC calcification (vitamin K1 −21.4%; Figure 4A). Conversely, serum from the placebo group did not have the same response, whereas in fact an increase in in vitro hVSMC calcification was found (placebo +73.2%; Figure 4A). There was a trend in hVSMC calcification between 12 months of vitamin K1 and placebo supplementation (Figure 4A; *p* = 0.06).

Next, we assessed dp-ucMGP levels between the two groups, as dp-ucMGP is both a vitamin K-dependent protein and a circulating marker associated with the development of vascular calcification (Figure 4B). Dp-ucMGP plasma levels were significantly decreased in the serum of individuals who received vitamin K1 supplementation (*p* < 0.0001). Comparing the delta in Agatston scores (change in calcification between pre- vs. post-supplementation), no differences between vitamin K1 or placebo could be observed (Figure 4C).

## 4. Discussion

Vascular calcification independently predicts the risk of cardiovascular disease [4]. Vascular calcification can be measured clinically by imaging techniques such as computed tomography, intravascular ultrasound and magnetic resonance imaging. These techniques are expensive and place burdens on patients. Measurement of circulating biomarkers can be a more cost-effective and less burdensome alternative for diagnosis of vascular calcification. However, to date, no such biomarkers have been identified that adequately reflect the calcification burden.

This paper describes the BioHybrid assay for determining the vascular calcification propensity of an individual by measuring the calcification response of cultured hVSMCs brought into contact with serum or plasma of that individual. The BioHybrid assay is built on the idea that the circulating biomarker of vascular calcification is not a single biomarker but a collection of circulating components. These can be sensed, integrated, and converted into a measurable calcification response by cultured hVSMCs. We describe the setup of the BioHybrid assay and its first validation.

Two methods are widely used for measurement of precipitated calcium salts in cell culture in vitro. These are Alizarin Red staining and o-Cresolphthalein quantification, and both are endpoint measurements. In order to be able to retrieve kinetics of calcification in the presence of hVSMCs, the BioHybrid assay employs fluorescently labelled fetuin-A in combination with live cell imaging. Fetuin-A is a protein abundantly present in the blood with the ability to bind to minerals with high affinity [10]. The amount of calcification measured with fetuin-A-AlexaFluor^®^-546 significantly correlated with the results obtained with the o-Cresolphthalein method (R^2^ = 0.72). During the early phase of in vitro hVSMC calcification, the fetuin-A-AlexaFluor^®^-546 probe is vastly superior with a greater degree of sensitivity than o-Cresolphthalein. Additionally, end-point assays require multiple replicates to record calcification development over time, which also increases chances of variability. With the BioHybrid assay, we exploit live-cell imaging, enabling us to follow the calcification development of a single condition and its replicates over time. Trialling the BioHybrid assay with a variety of both human serum and plasma samples grants further efficacy to the use of the BioHybrid assay as well as confidence in its robustness.

To date, most measurements are based on single protein biomarkers. The T50 assay was the first attempt to more comprehensively analyse the calcification propensity test by measuring the transformation of CPPs [6,11]. The T50 determines poor cardiovascular prognosis and has been shown to be associated with CVD, all-cause mortality and aortic stiffening in renal disease [12,13,14] based on serum factors and time to form CPPs. However, this assay is solely based on a chemical reaction without considering the vascular influence.

After optimizing calcification conditions, we tested whether T50 results could be replicated in the BioHybrid, comparing serum samples from pre- and post-dialysis. We found that the BioHybrid assay could measure differences between pre- and post-dialysis serum. Additionally, the BioHybrid could also discriminate between post-dialysis and control serum, i.e., a discrimination that the T50 assay was unable to make. The comparison between pre- and post-dialysis was performed since dialysis removes circulating components involved in calcification. The T50 is based on systemic proteins (pro/anticalcification) and minerals (i.e., phosphate, calcium and magnesium). However, factors impacting vascular cells that are involved in the calcification process (i.e, hVSMCs) are not measured in the T50. In contrast to the T50, these factors (i.e., toxins, cytokines) are also measured with the BioHybrid assay. Thus, the BioHybrid is measuring the sum of all factors, including vascular response, contributing to vascular calcification. Following the samples in real-time, six days in culture revealed a greater difference between pre- and post-dialysis as well as between post-dialysis and control. In addition to this, the ability to follow the same sample live and its replicates in real-time allows us to find specific responses as to the rate of calcification development. The differential response is possibly representative of the varying profiles exhibited by individuals on dialysis and requires further investigation. We suggest that as a next step, a more clinically coupled application of the BioHybrid in dialysis could serve as a warning or signal to non-response to treatments.

Given our application of the BioHybrid assay to the serum of CKD5D patients, we next checked how serum or plasma samples from a variety of cohorts would respond. This was done with serum or plasma samples from cohort participants with CKD stages 3–5, PXE and high vs. low Agatston scores.

PXE is a monogenetic liver disorder, and reduced plasma anti-mineralisation capacity is observed along with extensive ectopic calcification. As PXE is a metabolic disorder, assessing whether the blood-metabolites produced by the liver cause ectopic mineralisation makes it ideal for our BioHybrid screening [15,16]. We found that PXE plasma samples had a significantly increased effect on in vitro hVSMC calcification when compared to plasma from spouse controls. This suggests a circulating factor causing increased in vitro ectopic calcification. Further research is required to define the factors that are involved in PXE mediated vascular calcification, but our research indicates that the BioHybrid platform can distinguish factors present in the blood that affect calcification. Further, the BioHybrid could serve as a screening platform of drugs that target PXE induced mineralisation.

Severe renal artery and aortic calcification, CAC and AVC, are often symptomatic in CKD [17]. Additionally, up to 93% of CKD individuals on dialysis display imageable vascular calcification [18,19]. The Agatston score is evaluated as the gold-standard clinical predictor for the accumulation and progression of vascular calcification over time [5,20]. Further, CAC score is used to identify non-CKD high-risk individuals that need immediate medical attention or intervention [21]. In the BioHybrid assay, we found that all cohort patients (CKD stages 3–5, PXE and high Agatston scoring) showed increased in vitro calcification compared to respective controls. In vitro calcification progression with CKD5 serum was faster and greater than that of serum from CKD 3 and 4 patients and non-CKD controls. Serum samples from CKD 4 had the second greatest induction, followed by CKD3 serum and lastly control serum. This is somewhat representative of the clinical situation wherein an increase of severity of CKD is associated with an increase in vascular calcification [22,23,24].

Next, we tested whether serum from high and low Agatston scored individuals would have an influence in the BioHybrid assay. We found that hVSMCs calcification with serum from the high Agatston score group developed more calcification compared to the low Agatston score group. We further observed a positive correlation of calcification development within the high Agatston score group which was non-existing in patients with low Agatston score. Lastly, we tested serum samples from a cohort of participants with CVD diagnosis who had received either vitamin K1 supplementation or placebo for 12 months. Interestingly, serum from the vitamin K1 supplementation group had a reduced effect on in vitro hVSMC calcification compared to serum from the placebo group following 12 months of supplementation. The effect was most likely caused by changes in serum composition induced by vitamin K1, which is known to act as cofactor to activate vitamin K-dependent proteins (VKDPs) involved in the inhibition of calcification. This was not a direct effect of vitamin K serum levels as exogenous addition of vitamin K1 to serum had no effect on hVSMC calcification. We postulate that the protective effect of vitamin K1 in our BioHybrid assay is based on post-translational modifications of VKDP.

Although in the vitamin K1 study, all groups’ cardiovascular status continued to decline, the placebo group had on average a 25% increase in Agatston scoring during the 12-month period, whereas the vitamin K1 group had only a 12% increase [7]. Combined, these results indicate that the BioHybrid assay might have the potential to predict in vivo vascular calcification development and has potential sensitivity to distinguish amongst highly susceptible at-risk individuals.

Serum levels of the vitamin K-dependent protein and circulatory biomarker dp-ucMGP positively associates with vascular calcification [4,25]. Vitamin K supplementation decreased dp-ucMGP levels in all samples, yet this was not reflected in the clinical outcome for development of vascular calcification. We reason that dp-ucMGP is only reflective as a single biomarker involved in vascular calcification, a multifactorial biological process. Whereas dp-ucMGP reflects the status of a singular protein in circulation, the BioHybrid assay employs the whole blood compartment. We hypothesise that the BioHybrid platform can become an assay for clinical risk assessment of vascular calcification progression.

## 5. Conclusions

In conclusion, we present a novel BioHybrid assay that can determine personal vascular calcification propensity. Using fetuin-A-AlexaFluor^®^-546, we developed a real-time in vitro calcification assay that can give a quantifiable readout of in vitro hVSMC calcification development over time. We showed that vascular calcification is the consequence of the collection of all blood components that can be sensed and integrated into a calcification response by human vascular smooth muscle cells (hVSMCs). Further, the sensitivity of this assay has been demonstrated in response to dialysis, vitamin K treatment, as well as both metabolic and non-metabolic disorders that directly affect cardiovascular status. We propose wide scale application of this assay in larger cohorts to further validate the potential application of the BioHybrid assay as a non-invasive cardiovascular diagnostic tool, which may ultimately add to personalised risk assessment for CVD (Figure 5).

## Figures and Tables

**Figure 1 cells-10-02097-f001:**
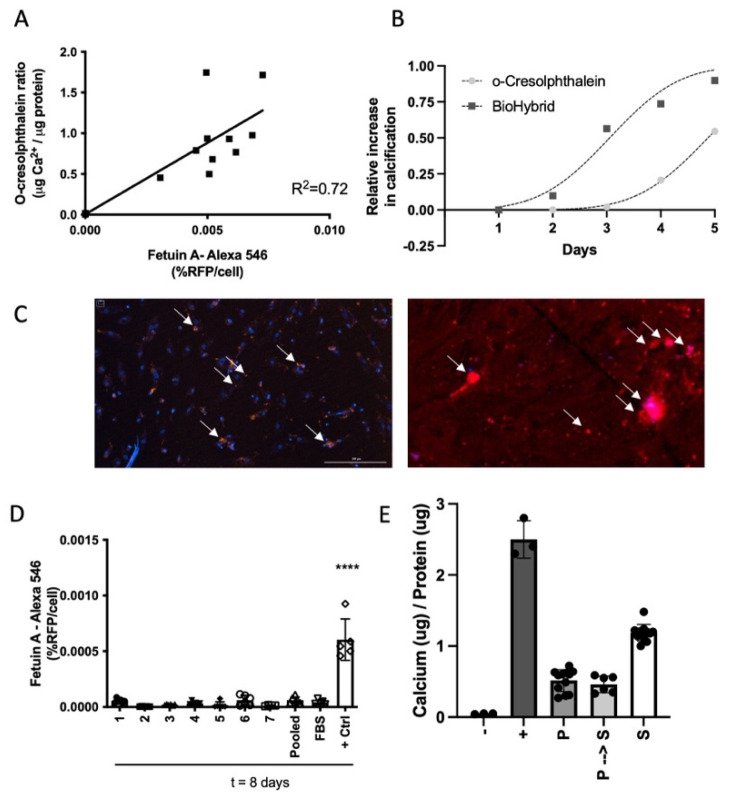
BioHybrid calcification assay. (**A**) Live calcification assay compared to the o-Cresolphthalein method provides a good correlation, R^2^ = 0.72. (**B**) Relative increase in calcification of both BioHybrid and o-Cresolphthalein methods. (**C**) Representative images (4× magnification) of the BioHybrid calcification assay for minimal and severe calcification. (**D**) Calcification assay comparing FBS to serum of healthy individuals (nr. 1–7) and control serum from healthy donors (pooled; *n* = 7). (**E**) Calcification of control plasma vs. serum (P = plasma (*n* = 4); P → S = recalcified plasma (*n* = 4); S = Serum (*n* = 4)). Positive controls are always under 0.5% serum conditions. **** *p* < 0.0001.

**Figure 2 cells-10-02097-f002:**
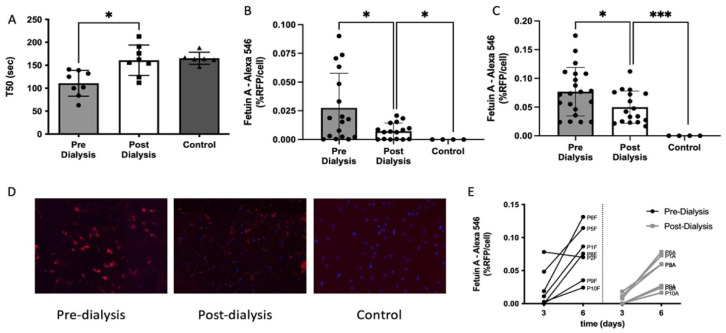
hVSMC calcification of CKD5D serum. (**A**) Serum from pre- (*n* = 8) and post-dialysis (*n* = 8) as well as control samples (*n* = 6) were tested in the T50 assay; a significant difference was found between dialysis, but no difference between post-dialysis and control serum. (**B**) Effect of dialysis on hVSMC calcification at day 3. A significant increase in calcification was found in serum of pre-dialysis (*n* = 7) vs. post-dialysis (*n* = 7). (**C**) Effect of dialysis on hVSMC calcification at day 6. A significant increase in calcification was found in serum pre-dialysis (*n* = 7) vs. post-dialysis (*n* = 7). (**D**) Representative images (4× magnification) of the calcification assay, control vs. pre- and post-dialysis CKD5D serum. (**E**) The effect of dialysis on the in vitro calcification rate. Serum of pre-dialysis has a significant higher calcification rate as compared to the same individual post-dialysis. * *p* < 0.05; *** *p* < 0.001.

**Figure 3 cells-10-02097-f003:**
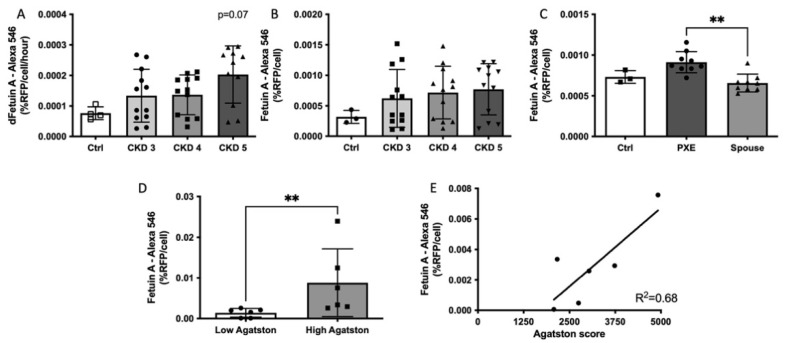
hVSMC calcification propensity of serum or plasma from cohorts of CKD stages 3–5, PXE and high vs. low Agatston score. (**A**) Calcification rate in the BioHybrid calcification assay. A trend was found in serum from CKD patients (*n* = 4) compared to control. (**B**) Calcification signal after 7 days in BioHybrid assay showing an increase in calcification when challenged with CKD serum (*n* = 4). (**C**) Calcification of PXE plasma (*n* = 3) compared to their spouses (*n* = 3) and healthy controls. A significant increase in in vitro calcification was found in PXE plasma compared to their spouses (*p* = 0.0012). (**D**) hVSMC calcification of serum from low (*n* = 6) vs. high (*n* = 6) Agatston score groups. A significant increase in calcification was found from serum of high Agatston scoring group compared to the low Agatston score group (*p* = 0.0022). (**E**) Correlation of Agatston score vs. in vitro calcification. A significant correlation was found in the high Agatston score group (R^2^ = 0.68; *p* = 0.0436) but not in the low Agatston group. ** *p* < 0.01.

**Figure 4 cells-10-02097-f004:**
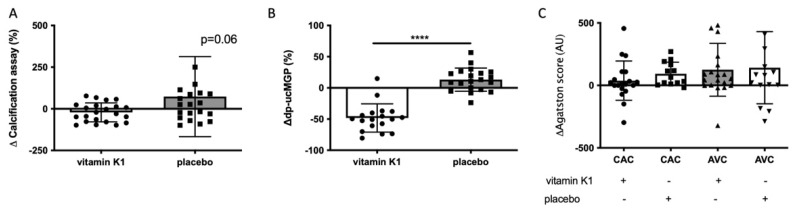
hVSMC calcification induction from serum of AVC and CAC patients. (**A**) In vitro calcification after 1 year vitamin K1 supplementation (*n* = 22) or placebo (*n* = 20). Supplementation of vitamin K1 showed a decreased response to in vitro calcification development comparing baseline serum to 12 months. Conversely, the placebo group exhibited increased in vitro calcification development in our assay (*p* = 0.06). (**B**) Dp-ucMGP change after 1 year. Dp-ucMGP levels significantly decreased following 1 year of vitamin K1 supplementation (*n* = 18) compared to the placebo (*n* = 20) group (*p* < 0.0001). (**C**) Development of Agatston score. No differences were found in either CAC or AVC score in both the vitamin K1 (CAC *n* = 18; AVK *n* = 19) and placebo (CAC *n* = 13; AVK *n* = 16) group. **** *p* < 0.0001.

**Figure 5 cells-10-02097-f005:**
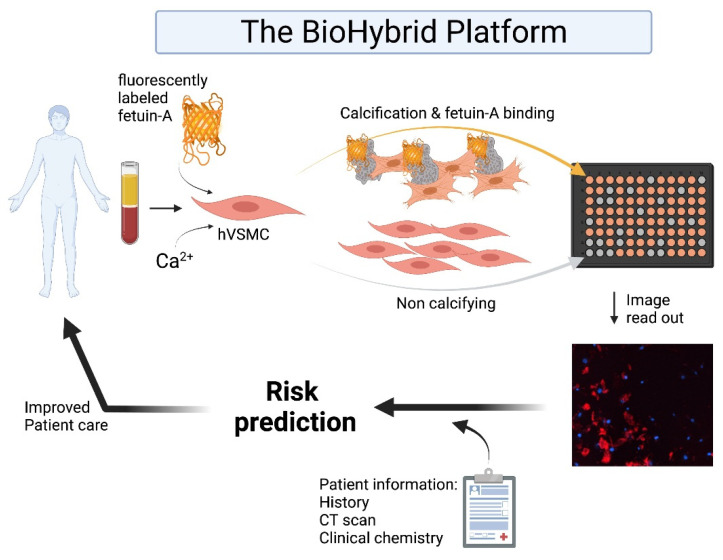
The BioHybrid platform. hVSMC mediated calcification is measured by fluorescent imaging using fetuin-A-AlexaFluor^®^-546. Created with BioRender.com.

## Data Availability

Data relating to this article can be found in the article itself or online Appendix A. Other material pertaining to this manuscript is available from the corresponding author pending reasonable request.

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
