# Peer review of "Development of the BioHybrid Assay: Combining Primary Human Vascular Smooth Muscle Cells and Blood to Measure Vascular Calcification Propensity"

_cells, 2021, doi:10.3390/cells10082097_

Round 1

Reviewer 1 Report

The most important sentence can be found in the discussion: The BioHybrid assay …. by cultured hVSMCs. (line 297-300). Indeed, more components in the blood/serum are involved in the process of vascular calcification, and therefore one needs an assay that is capable to measure sum of all these components. Does the here proposed BioHybrid assay this? I think that the authors have added a valuable assay to calcification diagnosis. Having said that, I would like to add some suggestions to improve this manuscript.

General remarks:
1) The authors have tested the BioHybrid assay with a variety of patients that are known to have a propensity for vascular calcification. To that end they incubated hVSMCs with sera or plasma of such patients in the presence of Fetuin A-Alexa 546. The authors claim that the BioHybrid method provides a dynamic representation of the calcification process (line 34). However, in the result section hardly any data on the dynamics of the assay are presented. Fig.1B does not tell a lot on the dynamics. The authors must have data on time course under different conditions. To be able to claim that the BioHybrid assay is superior to the other two assays, a direct comparison between samples tested would make the study more convincing. A little like the comparison in Fig.2A,B,C, but then more elaborated/detailed. For instance, in a table the results for the different tests and the Agaston score for the different patient groups or even individuals, could be presented.
2) The description of the Biohybrid method is, in my opinion too concise. The authors compared the results of the BioHybrid assay with those of the O-Cres method and TC50. The description of both methods is whether extremely (O-Cres) short or absent (TC50) in the M&M section.
3) The assay is based on the property of hVSMCs to accumulate calcium deposits under certain conditions. Because human primary cells are used, the batches of these cells can have different characteristics (even if they are characterized by the immunological markers mentioned in section 2.1). Even more so, because, in culture, VSMCs have a tendency to dedifferentiate gradually from a contractile phenotype to a synthetic/proliferative phenotype. So, it will be a mixed population one is working with, different for each laboratory that is using the BioHybrid assay. Therefore, the authors should elaborate on the stability and uniformity of the hVSMCs, and confirm/proof that despite possible variability, the BioHybrid assay superior is to endpoint assays (line 311).
Besides, I wonder whether it is necessary to work with human cells? Are established VSMC lines not a more uniform alternative. Did the authors test cell lines such as A7r5 or PAC cells, not human but stable over time.
4) Taking in account that the authors hypothesize that a variety of blood components contribute to calcification (line 25; 297), I wonder what the rational is behind the comparison between pre- and post-dialysis serum is (Fig.2). The dialysis has a marked effect on both T50 as well as BioHybrid assay. Likely some blood components are removed by dialysis, surprisingly, with different effects in the two assays. The discussion dealing with this is incomplete. Since calcification is an accumulative process, it is not surprising that the outcome on day 6 is more pronounced than on day3. I am more puzzled by the difference between p- and post-dialysis.
5) Neither the Agaston score nor vitamin K are mentioned in the abstract, but rather prominent presented in the results and the discussion section.

Small stuff:
Line 29: TC50 assay; Line 221, 324, etc.: T50 assay (the same?)
Line 239, Fig.3C: What is the difference between the ‘spouses’ and controls? A circulating factor (line 345)? Eating the same food? Sleeping in the same bed? Must be a hormone!!

Author Response

The most important sentence can be found in the discussion: The BioHybrid assay …. by cultured hVSMCs. (line 297-300). Indeed, more components in the blood/serum are involved in the process of vascular calcification, and therefore one needs an assay that is capable to measure sum of all these components. Does the here proposed BioHybrid assay this? I think that the authors have added a valuable assay to calcification diagnosis. Having said that, I would like to add some suggestions to improve this manuscript.

General remarks:

 1) The authors have tested the BioHybrid assay with a variety of patients that are known to have a propensity for vascular calcification. To that end they incubated hVSMCs with sera or plasma of such patients in the presence of Fetuin A-Alexa 546. The authors claim that the BioHybrid method provides a dynamic representation of the calcification process (line 34). However, in the result section hardly any data on the dynamics of the assay are presented. Fig.1B does not tell a lot on the dynamics. The authors must have data on time course under different conditions. To be able to claim that the BioHybrid assay is superior to the other two assays, a direct comparison between samples tested would make the study more convincing. A little like the comparison in Fig.2A,B,C, but then more elaborated/detailed. For instance, in a table the results for the different tests and the Agaston score for the different patient groups or even individuals, could be presented. 

We agree with the reviewer that one of the major advantages of the Biohybrid assay over existing assays is the ability to measure dynamically. The dynamic measurement of calcification allows us to choose the best time-point for calcification analysis. We have now replaced figure 1B and show the advantage of the BioHybrid method over the existing calcification methods. In Figure S1B we now show optimisation of the BioHybrid platform. By representing the BioHybrid in this way we can clearly show that we can examine kinetic courses of calcification and select time points that reveal differences in calcifying potential between patients. We believe that with changing these panels we provide a stronger message and thank the reviewer for this insight.

2) The description of the Biohybrid method is, in my opinion too concise. The authors compared the results of the BioHybrid assay with those of the O-Cres method and TC50. The description of both methods is whether extremely (O-Cres) short or absent (TC50) in the M&M section.

We thank the reviewer for raising this point. We have expanded the description of the o-Cresolphthalein method and added description of the T50 method.

3) The assay is based on the property of hVSMCs to accumulate calcium deposits under certain conditions. Because human primary cells are used, the batches of these cells can have different characteristics (even if they are characterized by the immunological markers mentioned in section 2.1). Even more so, because, in culture, VSMCs have a tendency to dedifferentiate gradually from a contractile phenotype to a synthetic/proliferative phenotype. So, it will be a mixed population one is working with, different for each laboratory that is using the BioHybrid assay. Therefore, the authors should elaborate on the stability and uniformity of the hVSMCs, and confirm/proof that despite possible variability, the BioHybrid assay superior is to endpoint assays (line 311). 
Besides, I wonder whether it is necessary to work with human cells? Are established VSMC lines not a more uniform alternative. Did the authors test cell lines such as A7r5 or PAC cells, not human but stable over time.

We agree with the reviewer that hVSMCs constitute a determining factor that may vary between laboratories. Therefore, it is important to compare the calcification propensity between patients using one batch of VSMCs. We also agree with the reviewer that primary cells can be an extra source of variability. To limit the variability as much as possible we characterise the cells extensively at low passage number and use the cells until a predefined time in culture/passage number. This can be found in the method section (line 98). 

As to the reviewer’s remark about the requirement for human cells we expect that theoretically rat VSMCs for example can be used to test calcification propensity of samples. However, we do not know whether there are species specific interactions between circulating components and VSMCs that influence the calcification response. Therefore, we do not recommend to use VSMCs from other species to determine the calcification propensity of patient samples.

4) Taking in account that the authors hypothesize that a variety of blood components contribute to calcification (line 25; 297), I wonder what the rational is behind the comparison between pre- and post-dialysis serum is (Fig.2). The dialysis has a marked effect on both T50 as well as BioHybrid assay. Likely some blood components are removed by dialysis, surprisingly, with different effects in the two assays. The discussion dealing with this is incomplete. Since calcification is an accumulative process, it is not surprising that the outcome on day 6 is more pronounced than on day3. I am more puzzled by the difference between p- and post-dialysis. 

We thank the reviewer for this comment. We concur with his/her reasoning and have added our rationale to make a comparison between pre- and post-dialysis serum. We further added an interpretation of our findings from the perspective of the BioHybrid concept at line 340:

“The comparison between pre- and post-dialysis was performed since dialysis removes circulating components involved in calcification. The T50 is based on systemic proteins (pro/anticalcification) and minerals (i.e. phosphate, calcium and magnesium). However, factors impacting vascular cells that are involved in the calcification process (i.e VSMCs) are not measured in the T50. In contrast to the T50, these factors (i.e. toxins, cytokines) are also measured with the BioHybrid assay. Thus, the BioHybrid is measuring the sum of all factors, including vascular response, contributing to vascular calcification.”

5) Neither the Agaston score nor vitamin K are mentioned in the abstract, but rather prominent presented in the results and the discussion section. 

We added a sentence in the abstract to highlight the important of the clinical data (line 40).

“Finally, vitamin K1 supplementation showed lower in vitro calcification reflecting changes in delta Agatston scores. Lower progression within the BioHybrid and on Agatston scores was accompanied by lower dephosphorylated-uncarboxylated matrix Gla protein levels.”

Small stuff:

Line 29: TC50 assay; Line 221, 324, etc.: T50 assay (the same?)

The TC50 from line 29 is a typo. Thank you for seeing this we have now corrected it.

Line 239, Fig.3C: What is the difference between the ‘spouses’ and controls? A circulating factor (line 345)? Eating the same food? Sleeping in the same bed? Must be a hormone!!

Spouses are partners of the patients, and therefor offer the best option to rule out the effect of potentially influential factors, e.g. lifestyle. Spouses are known to not suffer from PXE and are thus the controls of PXE patients.

Reviewer 2 Report

This manuscript of Armand M.G. Jaminon et al. addressed to measure in vitro calcification by BioHybrid assay there they developed a new cell-based high-content assay. The BioAssay was compared with the O-Cresolphtalein assay and the TC50 assay. Serum and plasma were derived from different cohort studies including chronic kidney (CKD) stages III, IV, V and V dialysis, pseudoxanthoma (PXE) and other cardiovascular diseases including serum from participants with mild and extensive coronary artery calcification

They recent study found that the BioHybrid assay was more sensitive to measure calcification propensity than the T50 assay as determined by measuring control and pre-and post dialysisserum samples of CKD patients.

Comments:

  1. In figure (1E.) we can see the extracellular Ca content. Did you make Alizarin Red Staining? If you did not, I would like to suggest to make this kind of experiment.
  2. Elevated serum phopshate levels have been linked with vascular calcification and mortality among dialysis patients.Why did you use just CaCl2 to induce calcification?
  3. What was the reason to choose M199 medium to culture hVSMCs. Most of the cases to induce vascular calcification of hVSMCs researchers are using DMEM high glucose content medium.
  4. Hyperphophatemia, elevated calcium-phophorus product, hypocalcemia and vitamin D deficiency are prevalent in dialysis patients. Calciphylaxis has been traditionally considered as a manifestation of severely dysregulated calcium-phophorus metabolism in dialysis patients. It would be interesting to examine the serum or plasma samples of Calciphylaxis’s patients.
  5. I would suggest to check the „in vitro” phrase in lane 162, 163,172, 173, 176, 216,234,238,243,266,268,303,309,345,357,358.
  6. In lane 177, instead of rapid I suggest to use fast.

Author Response

This manuscript of Armand M.G. Jaminon et al. addressed to measure in vitro calcification by BioHybrid assay there they developed a new cell-based high-content assay. The BioAssay was compared with the O-Cresolphtalein assay and the TC50 assay. Serum and plasma were derived from different cohort studies including chronic kidney (CKD) stages III, IV, V and V dialysis, pseudoxanthoma (PXE) and other cardiovascular diseases including serum from participants with mild and extensive coronary artery calcification

They recent study found that the BioHybrid assay was more sensitive to measure calcification propensity than the T50 assay as determined by measuring control and pre-and post dialysis serum samples of CKD patients.

Comments:

  1. In figure (1E.) we can see the extracellular Ca content. Did you make Alizarin Red Staining? If you did not, I would like to suggest to make this kind of experiment.

We thank the reviewer for this comment and agree that an Alizarin Red Staining would prove that the calcium we measure is extracellular. We have included an Alizarin Red image of our calcified hVSMCs which confirms that the extracellular calcium is stained red.

  1. Elevated serum phopshate levels have been linked with vascular calcification and mortality among dialysis patients. Why did you use just CaCl2 to induce calcification?

Indeed, increased phosphate levels are linked to vascular calcification and it is possible to induce calcification with phosphate in vitro. However, phosphate is also known to induce a VSMC phenotypic switch toward the osteogenic phenotype, a feature we did not desire in this version of our assay.

What was the reason to choose M199 medium to culture hVSMCs. Most of the cases to induce vascular calcification of hVSMCs researchers are using DMEM high glucose content medium.

We have choosen M199 to culture hVSMCs because we previously studied VSMCs induced calcification using VSMCs cultured in M199 (for example see Kapustin et al. Circulation Research 2015). Composition wise, both media are similar except for glucose levels. M199 contains 5.6 mM glucose and DMEM 25 mM. This difference in glucose levels is known to have minimal impact on VSMCs induced calcification in vitro.  

Hyperphophatemia, elevated calcium-phophorus product, hypocalcemia and vitamin D deficiency are prevalent in dialysis patients. Calciphylaxis has been traditionally considered as a manifestation of severely dysregulated calcium-phophorus metabolism in dialysis patients. It would be interesting to examine the serum or plasma samples of Calciphylaxis’s patients.

We concur with the reviewer that studying and challenging serum/ plasma of calciphylaxis patients would be highly interesting in our BioHybrid assay. Unfortunately, we did not obtain such samples yet.

I would suggest to check the „in vitro” phrase in lane 162, 163,172, 173, 176, 216,234,238,243,266,268,303,309,345,357,358.

We adjusted al “in vitro” to “in vitro”.

In lane 177, instead of rapid I suggest to use fast.

We adjusted "rapid" to "faster".

Reviewer 3 Report

The authors present a research article in which they developed an assay for detecting vascular calcification from patient serum or plasma. The method involves treating human smooth muscle cells with patient serum or plasma in the presence of Alexa-546 tagged Fatuin A. The authors compared head-to-head with other assays such as TC50 and O-Cresolphthalein. They mention some advantages of this technique over the others as being able to better discern between post-dialysis and control samples and potentially pick up severity of calcification in patients. Overall, the methods used in this study were appropriate and reasonable conclusions were drawn from the results. They do mention a wider study is required to further validate this assay.

My only comment is that authors should explicitly state their n’s for each experiment especially in Figs 2, 3 and 4. The dot plots are appreciated but it is easier for the reader understand the rigor of the experiments with the n’s in the legend.

Author Response

The authors present a research article in which they developed an assay for detecting vascular calcification from patient serum or plasma. The method involves treating human smooth muscle cells with patient serum or plasma in the presence of Alexa-546 tagged Fatuin A. The authors compared head-to-head with other assays such as TC50 and O-Cresolphthalein. They mention some advantages of this technique over the others as being able to better discern between post-dialysis and control samples and potentially pick up severity of calcification in patients. Overall, the methods used in this study were appropriate and reasonable conclusions were drawn from the results. They do mention a wider study is required to further validate this assay 

My only comment is that authors should explicitly state their n’s for each experiment especially in Figs 2, 3 and 4. The dot plots are appreciated but it is easier for the reader understand the rigor of the experiments with the n’s in the legend.

We thank the reviewer for the positive comments on our manuscript. Indeed, to further validate our study we need to include a wider range of patients and increase the number of replicates. We now added all the n’s for each experiment individually in the figure legends.